# Mobilisation towards formal employment in the healthcare system: A qualitative study of community health workers in South Africa

**Hlologelo Malatji**[1]*, **Frances Griffiths**[1,2], **Jane Goudge**[1]

**1** Centre for Health Policy, School of Public Health, University of the Witwatersrand, Johannesburg, South Africa, **2** Division of Health Sciences, Warwick Medical School, University of Warwick, Coventry, United Kingdom

\* Hlologelo.Malatji@wits.ac.za

**Data Availability Statement:** All data supporting our work is provided within the manuscript.

**Funding:** The study was supported by Medical Research Council UK, Department of International

## Abstract

In low and middle-income countries, community health workers (CHWs) play a critical role in delivering primary healthcare (PHC) services. However, they often receive low stipends, function without resources and have little bargaining power with which to demand better working conditions. Using a qualitative case study methodology, we studied CHWs' conditions of employment, their struggle for recognition as health workers, and their activities to establish labour representation in South Africa. Seven CHW teams located in semi-urban and rural areas of Gauteng and Mpumalanga Provinces were studied. We conducted 43 in-depth interviews, 10 focus groups and 6 observations to gather data from CHWs and their representatives, supervisors and PHC facility staff. The data was analysed using thematic analysis method. In the rural and semi-urban sites, the CHWs were poorly resourced and received meagre remuneration, their employment outsourced, without employment benefits and protection. As a result of these challenges, the CHWs in the semi-urban sites established a task team to represent them. They held meetings and caused disruptions in the health facilities. After numerous unsuccessful attempts to negotiate for improved conditions of employment, the CHWs joined a labour union in order to participate in the local Bargaining Council. Though they were not successful in getting the government to provide permanent employment, the union negotiated an increase in their stipend. After the study ended, during the height of COVID-19 in 2020, when the need for motivated and effective CHWs became more apparent to decision makers, the semi-urban-based teams received permanent employment with a better remuneration. The task team and their protests raised awareness of the plight of the CHWs, and joining a formal union enabled them to negotiate a modest salary increase. However, it was the emergency created by the world-wide COVID-19 pandemic that forced decision-makers to acknowledge their reliance on this community-based cadre.

Development (DFID), Economic and Social Research Council (ESRC) and Wellcome Trust under the Joint Health Systems Research Initiative (JHSRI) (Grant number: MR/NO15908) awarded to JG. It was also supported by the South African National Research Foundation (NRF) through the South African Research Chair (SARChi) in Health Systems and Policy awarded to the Centre for Health Policy, University of the Witwatersrand, and held by FG. All three authors received salary supplementation as part of the research funding from JHSRI. The funders were not involved in the study design, data collection, analysis and preparation of this manuscript.

**Competing interests:** The authors have declared that no competing interests exist.

## Introduction

In pursuit of universal health coverage (UHC), low and middle-income countries (LMICs) are investing in programmes to strengthen the delivery of primary health care services (PHC) to marginalised communities [1]. The community health worker (CHW) programmes gained prominence after the Alma Ata Conference of 1978, which prioritised the strengthening of PHC to provide care to marginalised communities [2]. Since this period, many LMICs have been making attempts to use CHWs to provide care [3]. In some settings, the CHWs have been successful in extending access to PHC services by supporting chronic care management, antenatal and postnatal support and early identification of malnourished children [3–5].

However, in some settings, the programmes fail to deliver on expectations due to limited access to resources (e.g. blood pressure and glucose monitoring machines), low and sometimes irregular remuneration and poor integration into the healthcare system resulting in poor relationships with facility-based staff members [4, 6, 7]. CHWs are mostly women, often with low levels of education, who are expected to work on a voluntary basis, or who are underpaid with no permanent employment security [8]. These employment conditions and the lack of unionisation contribute to CHWs' low work morale, demotivation and high work turnover [1, 9]. Pandya et al argue the provision of incentives, both monetary and non-monetary is crucial to improve CHWs motivation, job satisfaction, and performance, which, in turn, improves retention [8]. Monetary incentives include stable salaries, while non-monetary incentives include the provision of working tools such as uniforms and raincoats.

While the World Health Organisation (WHO) has argued for the optimisation of CHW programmes through fair remuneration schemes and provision of career development pathways, as well as integration into health systems and provision of resources [7], there has been little progress. Unionisation among the CHWs is often low due to CHWs working in isolation or in pairs, with little bargaining power with which to demand better working conditions [10, 11].

In South Africa labour unions have a played a key role in the country's history in opposing Apartheid; the white minority government had enacted labour laws based on race categorisation thus resulting in unfair labour practices [12]. In the 1990s, the Congress of South African Trade Unions (COSATU), an umbrella body of unions, formed an alliance with the governing political party–the African National Congress [13]. As a result, unions participated in government policy discussions, and influenced labour legislation and strengthened employees' rights in the workplace. Public sector unions' alliance with government enabled the formally employed workers to consistently obtain above-inflation salary increases through centralised bargaining, although union leaders are often criticized for using the alliance to build their own political connections, rather than representing the concerns of ordinary workers [14, 15].

In line with international UHC goals, in 2011, the South African Department of Health (DOH) has attempted to strengthen the delivery of PHC services through various streams of care including the introduction of a nation-wide CHW programme (locally known as ward-based outreach teams WBOTs)) [16]. The programme intention was for each WBOT, linked to a local PHC facility, to comprise of 6 CHWs who serve a defined geographic area, provide a combination of promotive and preventative healthcare services to households, and make referrals to health facilities and social services organisations [16]. The team was to be led by a nurse and supported by a health promoter and an environmental officer.

Prior to the introduction of the WBOTs, the CHWs were in the employment of non-government organisations (NGOs), mainly providing HIV/AIDS and TB related services at community level [10]. The NGO programmes were fragmented and single disease-focused, thus neglecting the other needs of individuals and families. The WBOTs replaced the NGO-led

programmes and absorbed the CHWs on a 12-month renewable contract earning a monthly stipend of R2 500 (136 USD) (at the time of study), their employment was outsourced to either NGOs or payroll administration company. As a result, despite their employment within a nationwide CHW programme, the CHWs had little employment security and were paid a meagre stipend. With high levels of union activity being common in South Africa, informal groupings of CHW have been protesting for several years, demanding better working conditions [11, 17, 18]. In this article, we examine the CHWs' conditions of employment, their struggle for recognition as health workers, and their activities to establish labour representation in South Africa.

## Methods

### Study design

The data originate from a large project and a doctoral study that investigated the design and implementation of CHW programmes in South Africa. The larger project which was funded included the introduction of a nurse mentor intervention with a specific focus on building CHWs' knowledge and skills, supporting their integration into healthcare systems and community structures [19, 20]. Together, the two studies adopted a qualitative approach with 7 case studies (each case study being a team of CHWs). Creswell and Clark define qualitative research as an approach used by researchers to gain an in-depth understanding of a particular phenomenon [21]. Through this approach, the researchers undertook an in-depth exploration of the CHWs conditions of employment and mobilisation activities for recognition in the healthcare system.

### Study setting

The research was carried out in 3 health districts of Gauteng Province and 1 health district of Mpumalanga Province (Table 1). We applied the maximum variation technique to select the two provinces based on their location (urban vs rural), history and contrasting models of implementing the CHW programmes. The CHW programmes in the three districts of Gauteng province were implemented by the Provincial Department of Health with a payroll administration company contracted to pay the CHWs on behalf of the Department [17]. The Mpumalanga province, where the rural case studies were located, had not fully implemented the national CHW programme [17]. Here, the Provincial Department of Health funded existing local NGOs that provided home-based care services, to deploy some of its carers to

**Table 1. Data collection methods and number of participants.**

| | District A* | | District B* | District C* | | District D* | | Total |
|---|---|---|---|---|---|---|---|---|
| | Team 1 | Team 2 | Team 3 | Team 4 | Team 5 | Team 6 | Team 7 | |
| Focus group discussions with CHWs | 1 | 1 | 2 | 2 | 2 | 1 | 1 | **10** |
| Interviews with CHW team leaders | 4 | 4 | 3 | 2 | 1 | 1 | 2 | **17** |
| Interviews with nurse mentor | 4 | - | - | | - | | | **4** |
| Interviews with task team members/ CHW representatives | 6 | - | - | - | - | - | | **6** |
| Interviews with key informants | 5 | | 2 | 6 | | 3 | | **16** |
| Observations of task team meetings | 6 (meetings) | | - | - | | - | | **6** |
| Reports reviewed | 5 | | | | | | | **5** |

*Since some events being reported are contextual information. The authors have elected not to use the districts' real names in order to protect the identities of the participants.

constitute the CHW programme in local health facilities [17]. The NGOs remained the primary employers of the CHWs, while supervision and resources were being provided by the health facilities [17]. Though we had an interest in the two provinces, we spent 3 years in one district of Gauteng province. In the other districts, we couldn't stay long due to limited resources.

### Sampling and participant recruitment

We employed a combination of purposive and snowball sampling techniques to select the participants for the study [11]. All CHWs and supervisors who formed part of the WBOT/CHW programmes and facility staff members in the PHC facilities were purposively selected to participate in the study. The snowball sampling technique was used to select key informants in government and universities in Gauteng and Mpumalanga provinces (see Table 1). The informants' inclusion criteria was a sustained knowledge of CHW programmes, and involvement in the set-up, implementation and evaluation of the programmes performance in the different sites.

### Data collection

The data was collected by the first author and a group of data collectors from 12 September 2016 to 30 May 2019. The first author was a South African Research Chairs Initiative (SARChI) doctoral researcher employed by a research centre undertaking health policy and systems research. The data collectors were qualified qualitative researchers sensitised to the type of activities CHWs undertake. Focus group discussions, individual interviews and observations were used to collect data from the different CHWs, facility staff members and key informants. The research instruments were all in English, during interviews the data collectors translated the interview questions to local languages (i.e. IsiZulu, Sesotho and Sepulana) for participants who did not understand the English language.

**Focus group discussions.** Ten focus group discussions (FGDs) were held in the different sites, and each FGD had approximately 10 participants, all women (Table 1). All CHWs who formed part of the CHW programme in the PHC facilities were invited to participate in the FGDs, and there were no refusals. We used a structured guide to ask the CHWs about their work conditions, daily activities, resources they require to perform their duties and their engagement with the health facilities and their employer (Department of Health). Each of the discussions was audio-recorded and lasted approximately 2 hours. The first author facilitated the discussions while the data collectors took notes. The notes were typed by the data collectors and formed part of the dataset for analysis.

**Observations.** The observations were conducted to understand the CHW activities relating to labour mobilisation aimed at challenging the employer to provide better conditions of employment. With the consent of the CHWs, we observed 6 CHW representatives' meetings with the general CHW population, programme coordinators and the payroll administration company in District A (Table 1). In these meetings, we documented the issues being discussed (e.g. poor pay and lack of working tools) and resolutions made.

**Interviews.** In each PHC facility, we interviewed CHWs and their supervisors, facility staff members and nurse mentor (Table 1). The interviews happened after the observations of the CHW meetings, in order to get clarity on the CHWs' activities. We asked these participants about the CHW programme (i.e. CHWs duties in households, access to resources and supervision, and conditions of employment). Key informants at provincial and district levels were also interviewed.

Furthermore, in District A where we spent a prolonged time in the field, we interviewed CHWs who represented their colleagues at district level meetings and when there were

conflicts. The CHW representatives were referred to as a "task team". We asked them about the origin of the task team, engagement with health facilities, district and province, and successes and challenges in advocating for CHWs rights. Each interview lasted approximately 45 minutes and were audio-recorded with participant consent.

### Data analysis

As described by Braun [22], we used the thematic analysis method which was inductive to analyse the individual interviews, focus group discussions and participant observation data. The team read through the interview transcripts and observation notes and developed a codebook. The first author HM coded the interview transcripts and observation notes line-by-line, and to ensure the rigour of the findings the coded data and emerging themes were presented to the co-authors (JG and FG). The co-authors verified the coded data against the transcripts. The codes and supporting quotations captured information about the CHWs conditions of employment and activism to challenge the government to provide permanent employment with decent remuneration.

### Ethics approval and participant consent

The larger project was cleared by the University of the Witwatersrand HREC Medical Committee (M160354). The doctoral study received ethical clearance (M180540) from the same university ethics body and the research authorities in the 4 districts of Gauteng and Mpumalanga provinces. The participants provided written informed consent before participating in interviews and focus groups.

Before observing the CHWs and their representatives in meetings with district management, we asked those present for verbal consent. We explained there would be no audio-recording of the engagements; the researcher will write notes of the deliberations.

## Findings

The findings comprise of participants demographics, CHWs conditions of employment (i.e. stipend, employment security, career advancement opportunities, work tools and access to supervision), and how these conditions hindered the effectiveness of the CHWs in the different sites. Lastly, the CHW labour mobilisation towards permanent employment with decent remuneration is presented.

### CHW, supervisor and key informants' demographics

We studied 7 CHW teams in the health districts of Mpumalanga and Gauteng provinces. The teams' composition ranged from 8 to 25 CHWs per team. The youngest CHW was aged 26 while the oldest was 61 years old. The CHWs also had varying years of service ranging from 3 years to 17 years. The CHW teams had different supervision configurations consisting of professional nurses and enrolled nurses. The CHWs in Districts A and C were being supervised by professional nurses and enrolled nurses. The supervision arrangement was different in District D, where the CHWs were being supervised by professional nurses, while those in District B were being supervised by enrolled nurses. The supervisors were a diverse group aged between 29 years to 65 years. Before joining the government CHW programme, they had worked in different health institutions for 4 to 39 years. The key informants occupied positions such as programme coordinator and researcher in government departments and universities.

## CHWs conditions of employment

**Stipend.** Across the sites, the CHWs received a monthly stipend of R2 500 (135 USD). The CHWs said that the stipend was insufficient and demotivating them: "*. . .the government should hear us and increase our stipend, so when we go to the community to motivate and counsel people, they understand us. We cannot talk to people who are frustrated and hungry when ourselves are also frustrated and hungry*." (CHW, FGD 1, District C, Team 2). The CHWs felt undervalued: "*We become de-motivated because of the stipend. Our stipend is too little, and you see the amount of work that we do. We get burned by the sun*" (CHW, FGD, District C, Team 2). The CHWs provided services to patients located in informal settlements, hostels and remote areas. Some of the areas, particularly the hostels and informal settlements were unsafe due to high crime rates.

The CHWs also experienced difficulties in receiving their monthly stipend. In the urban sites, the Provincial Department of Health contracted a payroll administration company to pay the CHWs (Table 2). The company issued a bank card that the CHWs used to access their stipend, however, the card did not have features of a traditional bank card (e.g., cash deposit and transfer options); it was only to receive and withdraw the stipend. The company offices were located far from the reach of many CHWs, who lived and worked in the periphery of the districts, making it difficult to resolve any problems with the card or the stipend.

In the rural sites, where the programme was managed by local NGOs, the CHWs regularly received their stipend late, and would continue working for up to 6 months without being paid: "*Normally, we are getting paid after 3 or 6 months so now we might get paid in October/ November. This is disturbing us. We are always in debt and when we get that amount even if it is back pay it all goes to the creditors*" (CHW, FGD, District D, Team 2). The sub-district officials were aware of the delayed payment of CHW stipend and expressed how it is important for the CHWs to be formal employees of the department, so they can be paid on time: "*The CHWs receive their stipends from the home-based care organizations (NGOs). The minister mentioned that the CHWs should be part of the department and have PERSAL numbers but due to budget constraints, I don't see this happening soon*" (Government official, Interview, District D). The government official blamed the NGOs for the delayed payment of the CHWs as they always submit paperwork required by the Provincial Department of Health to release funds late.

**Table 2. Description of CHW programmes and labour representation.**

| | District A (semi-urban) | District B (semi-urban) | District C (semi-urban) | District C (rural) |
|---|---|---|---|---|
| **Programme management** | Government | | | Government; contracted to local NGOs |
| **Salaries disbursement** | Paid by a payroll administration company contracted by the government | | | Paid by NGOs contracted by the government; delays in disbursing salaries |
| **Access to resources** | The CHWs lacked basic working resources and tools such as stationary, equipment and office space to store patients' records and receive supervision | | | |
| **Member of union/ advocacy group** | Yes; prior to joining a formal labour union, the CHWs were represented by a task team | Yes | Yes | No |
| **Labour mobilisation activities** | • Regular protests to demand permanent employment<br>• Facility stay aways | Protests to demand permanent employment | Protests to demand permanent employment | None |
| **Outcomes of task team and union mobilisation** | • Monthly stipend increased<br>• Payroll administration company contract not renewed<br>• In District A, the task team negotiated with health facilities to attend to CHWs' needs (e.g. resources) | | | Monthly stipend increased |

**Employment security.** The urban-based CHWs were on 12 months fixed contracts and had to renew the contracts with the payroll administration company each year. This created anxiety among the CHWs: "*What if they wake up and say that they are not renewing our contracts? What are we going to do? We have children and families, the children are waiting on us as their moms to bring them something*" (CHW, FGD, District B). The CHWs also felt the company, as contractor to the Department of Health, was disinterested in their call for permanent employment: "*I can tell you that the payroll administration company doesn't care about us. If our task team says that it has a meeting, the payroll company does not understand or give us the go ahead to attend the meeting. We have to struggle to go to that meeting.*" (CHW, FGD, District A, Team 1).

The CHWs in rural sites were anxious about their contractual arrangement with the NGOs, as their employment was decided by the NGO managers. During our discussions, they seemed unable to openly discuss their employment concerns, as they feared being recalled from the government programme (where there was a possibility of permanent employment) by the NGO managers. The NGO managers had the power to deploy and recall the CHWs from the government CHW programme. These concerns often emerged after the audio-recording had stopped.

**Career advancement opportunities.** Some CHWs were frustrated by the lack of career progression opportunities in the field, as they wanted to establish a career in health care (e.g., to be trained as nurses). The CHW team in District B had young CHWs with matric qualifications who were interested in furthering their studies. They felt the Department of Health was not making opportunities available for them: "*In our team, there are people with mathematics and physical science; why can't they take these people so that they could study for something like nursing, pharmacy and so forth? They can see that we love what we are doing. Not all of us have the money to go to private institutions to study. The government is failing us*" (CHW, FGD, District B). One of the CHWs was already studying towards a teaching qualification at a local university. This was not true for the other districts, where the majority of the CHWs did not possess a matric pass, which is a prerequisite to enrol for tertiary education. However, there was still a desire among the CHWs to be trained and promoted into better positions with benefits.

A representative of the CHWs in District A was concerned that the Department of Health provides short training courses but never considers them (CHWs) for promotion after the training: "*After some of them finished and got their certificates; they are still doing the same work, same level, same stipend. That is our biggest challenge. Why are there trainings that don't have opportunities?*" (CHW representative, Interview 1, District A). This view was shared by the CHWs: "*We complete trainings. At least when you have done the training, they should promote you but no we remain in the same posts.*" (CHW, FGD, District B). It appears there was no career advancement opportunities for the CHWs even after completing training.

**Resources.** When undertaking household visits, CHWs require resources such as blood pressure and glucose monitoring machines, stationary and a raincoat etc. At the inception of the programme, the Department of Health made provisions for the CHWs to have these resources, although some teams did not receive them: "*We only saw the boxes. One sister from clinic X did ask how come we don't have BP machines. We said we don't know. She promised to get us BP machines. We got 2 machines and we had to share them but now there are no batteries*" (CHW, FGD, District D, Team 2). During data collection, the majority of the CHWs still did not have these resources (Table 2).

The lack of equipment affected the CHW activities in the households: "*Some patients expect us to check their blood pressure and sugar level when we visit them. If I don't have the machines, they ask what is the use of me visiting them?*" (CHW, FGD, District D, Team 6). Another CHW commented: "*The patients keep on asking us to measure their BP and blood sugar. . . we always tell them that we don't have equipment. That makes people to undermine us; they say we are useless.*" (CHW, FGD, District B).

A facility manager in District C was aware the CHWs needed resources, however, he felt helpless: "*They received backpacks, but the bags did not have all the equipment. The CHWs are supposed to have blood sugar machines but they don't have them. If the clinic were to distribute the glucose strips to all the CHWs, the clinic will be left with none and that wouldn't be good because they are very costly.*" (Facility Manager, Interview, District C, Team 4).

Space within the health facilities was an issue in both the urban and rural sites. A CHW commented: "*. . . sometimes we find that our things are missing or when you check your stuff you will find your things rearranged as if someone was looking for something, and you cannot ask anyone about it.*" (CHW, FGD, District C). The CHWs also bemoaned the lack of privacy while participating in capacity-building workshops: "*. . .we write pre-tests and post-tests weekly. When we write the tests, we need a quiet space to concentrate, but there will be facility staff members going back and forth while we are writing the test. While sitting here, another person would use the microwave*" (CHW, FGD, District C). In the rural sites, the CHWs did not have workspaces in the main facilities, instead they were accommodated in other buildings, which limited interactions and collaboration with facility-based staff members. However, teams with senior supervisors often had access to suitable spaces within the facility, as their supervisors were able to negotiate for them to use the spaces without being interrupted.

**Uniforms.** The government did not provide the CHWs with work uniform. The CHWs felt obliged to purchase the uniform, as they wanted the community to recognise them as members of the healthcare system. The CHWs felt if they visited the households dressed in private clothes, they will struggle to gain the trust and acceptance of the community. "*It is a must because you have to be presentable to the patients in the community, you cannot go to the community wearing a grey shirt, they will think you are a thief*" (CHW, FGD, District B). To achieve this end, they saved a portion of their stipend to buy the uniform. Some CHWs were unhappy to use their money to buy the uniform: "*The money is little but they still expect us to be presentable when we go to the households. We buy everything that we wear; they don't offer us anything like a t-shirt for work–nothing*" (CHW, FGD, District C, Team 2).

A facility manager in District C agreed that uniform plays a critical role in ensuring the CHWs are recognised as members of the local health facility: "*When someone looks presentable, they gain peoples' trust and attention easily. As long as the CHWs do not have uniform, name tags and continue receiving low stipends, the community will not take them seriously*" (Facility Manager, Interview, District C, Team 4).

**Relationship with facility-based staff members.** Across the different sites, the CHWs reported complex relationships with their supervisors, facility-based staff members and programme coordinators. Due to their low status in the health system, the CHWs felt neither appreciated nor respected by facility-based staff: "*They disrespect and undermine us but we do respect them, not that we are scared of them" (*CHW, FGD, District B). In some facilities, the CHWs who felt disrespected by facility staff resorted to physically fighting them as a way of asserting themselves (CHW, FGD, District B).

Despite poor relationships with co-workers, the CHWs continued to assist the health facilities in their daily functions. We observed the CHWs delivering long-term medications to the elderly, tracing medication defaulters and referring their patients to health facilities for further medical assistance. However, the CHWs felt facility staff members delayed attending to the patients they had referred to the facility, and this affected their ability to bring uncooperative patients for care. These issues were better managed in facilities where a senior supervisor was present. The senior supervisor responsible for Team 1 negotiated with facility-staff members to support the CHWs with timely attendance to referred cases. This arrangement motivated the CHWs and built their work morale.

A facility manager argued there is a need to integrate the CHW programmes into PHC facilities: "*They introduced WBOT as a programme as if it is independent from the clinic hence there isn't unity between the WBOT and facility staff. Even worse they employed managers for the WBOT programme. This implies the facility and WBOT are two different entities. So, when they go to the facility manager to ask for medication, the facility refuses and says that they are finishing their stock*" (Facility Manager, Interview, District C, Team 4).

**Supervision.**   There were visible differences in how supervision was provided at the different health facilities. In the urban sites, every morning before going into the community, the supervisor provided the CHWs with in-service training focusing on the cases they attended the previous day and accompanied them on household visits to provide support. In the rural sites, CHWs were managed by senior supervisors, but there were no morning meetings before going into the community, and they operated without community supervision as their supervisors were not able to go to the field with them, mainly due to the lack of transport.

Infrequent supervision was a concern for the CHWs and their representatives. In District A, the representatives complained that the supervisors were not investing sufficient time to attend to the needs of the CHWs, instead spent most of their time in the facilities: "*When they get in the clinic, they do clinical work, like children's immunizations, family planning, or whatever. They don't support the CHW programme, you understand? It is the challenge that we have, and you find when the CHWs tell the [CHW representatives] about this, it causes conflict with the supervisors*" (CHW Representative, Interview 2, District A). The representatives were unhappy that the supervisors spent most of their time in the health facilities while they were employed to oversee the CHWs activities in the community. The Department of Health, which is the employer of the supervisors, expect all CHW supervisors to spend 70% of their time in the community, 30% in facilities mainly performing administrative duties related to their functions.

## CHW labour mobilisation

As described above, The CHWs were frustrated by the lack of working tools, poor remuneration and absence of career advancement opportunities. In this section, we highlight the CHWs' activities to challenge the government to improve their conditions of employment.

**The emergence of CHW representation.**   Prior to joining the nation-wide CHW programme in 2011, the CHWs worked for NGOs providing home-based care services. While in the NGOs, the CHWs established a committee to challenge the exploitation perpetuated by the organisations' management (Table 2). After joining the national programme, representatives of the CHWs were elected to what became known as the task team. The task team is made up of CHWs, lay counsellors and health promoters working in different health facilities within the district. These CHW representatives decided not to become members of established labour unions, as existing unions were perceived as ineffective and with a history of siding with management. A member of the task team commented: "*We have tried to be under a labour union, but whenever they left to go to the negotiations, they did not take us. They don't even take one of us to be a part of the negotiations, even if the person is just an observer*" (CHW representative, Interview 1, District A). Another member added: "*The union that we will agree to be under will be a union that we will be a part of whenever they go to negotiate for us, or whatever. We must be included and we must be a part of that negotiation*" (CHW representative, Interview 1, District A).

The task team operating in District A ensured each of the 4 sub-districts had a representative: "*We have divided ourselves according to clinics; let us say he/she works at facility X. He/she is the one who is responsible for the facilities that are closer to her facility. I will also be*

*responsible for facilities closer to where I work. We divide ourselves like that*" (CHW representative, Interview 2, District A). The task team members are dependent on donations from CHWs to fund their travelling costs between the facilities.

In Districts B and C, there were also emerging groups of CHWs who advocated for the rights of the CHWs. In the rural sites, where the programme was under the management of NGOs, CHWs did not have formal representation. The CHWs seemed fearful of being outspoken about the need to have a labour union to represent them.

**Negotiating for permanent employment.** Since 2015, the CHW leaders have led multiple strikes and stay aways in Gauteng province. One strike was triggered by the appointment of the payroll administration company without consulting the CHWs. The CHWs were concerned that the provincial government was able to award a multi-million contract but never considered their demands for permanent employment: "*We wanted to understand why we were tendered without our consent? The department needs our services; why is it outsourcing us when it has money for tenders? That was our concern.*" (CHW representative, Interview 2, District A). Another member of the task team commented: "*We had a strike and spent about 3 days sleeping there. We came back and went back again to fight and spill rubbish in their offices, unfortunately, we fought without any luck*" (CHW representative, Interview 3, District A).

The task team leaders were arrested after embarking on this violent and unprotected strike (Non-procedural, or unprotected strike, is one where the strikers have not complied with the requirements of the South African Labour Relations Act of 1995 before going on strike. This removes the employer's opportunity to develop contingency plans for running the business during the strike.). During the strike and court appearances, the task team encouraged the CHWs to withdraw their services and to protest in support of their leaders at the court as their case was being heard. Some CHWs were not keen to participate, as it meant leaving their workplace and neglecting patients. Some CHWs who chose not to participate were threatened with violence. A facility staff member commented: "*They all donated a R10 for transport for those who will go and support the task team. The CHWs who wanted to work were receiving threats that they will come for them*" (Nurse Mentor, Interview 3, District A). Following the release of the task team members from jail, the CHWs were ordered to withdraw their services in celebration. A facility staff member commented: "*On Wednesday the 23rd, CHWs' didn't come to work because they were told by the task team to go on a 3 day holiday to celebrate their struggle and past imprisonment of their two task team leaders. There were no formal notification of this holiday and all WBOT leaders, including supervisors didn't know except district management*" (Nurse Mentor, Interview 2, District A).

After the unprotected strike action, the task team members continued efforts to improve CHWs conditions of employment. Within the health facilities, they focused on addressing CHW conflicts with facility staff members and lack of resources: "*We go to the facility and talk to the management and the CHWs about their issues. We request for a space for the CHWs, and if there is no space, the space must be created and they must have the space*" (CHW representative, Interview 2, District A). One of the accusations was that supervisors neglected the CHWs by not going to the community with them, the CHWs are left to fend for themselves while out in the community. The task team also ordered CHWs who were assisting facilities with administrative duties to withdraw their services: "*She stopped because the task team told her to stop. They must focus on being the CHWs. They are not receptionists*" (Nurse Mentor, Interview 4, District A).

Government officials in District A called a meeting aimed at resolving the tension between the CHWs and supervisors. At this meeting, the CHW representatives verbally abused the supervisors in the presence of the officials: "*Some of the [supervisors] were literally crying and some didn't want to talk. They were busy going up and down, going to the bathroom. They were*

*talking outside. I don't know what they were saying*" (Nurse Mentor, Interview 3, District A). Due to the confrontational approach adopted by the task team, the government officials didn't defend the supervisors or allow the supervisors to speak for themselves. As a result, the supervisors were left exposed at the meeting and the militant approach adopted by the task team created greater hostility between the parties. The nurse mentor reported: "*The task team has more influence than unions or district coordinators because they have control over the CHWs. For example, if they were to call the CHWs today and tell them not to go to work, the CHWs will do exactly that*" (Nurse Mentor, Interview 3, District A).

To resolve the conflicts, at the suggestion of a nurse mentor employed in two of the health facilities where we collected data, the task team was invited to a meeting to discuss areas of discomfort with supervisors and to work as partners. This engagement enabled the supervisors to be sensitive to the CHWs' attempts to seek improved conditions of employment. Post this meeting, the task team changed their approach to dealing with conflicts: "*When there is a concern, the person who complains is called together with the person that they are complaining about. We meet them in the same room. We listen to the complainant story, at the same time the defendant is present and is also listening and able to respond*" (CHW representative, Interview 2, District A).

In 2018, the Gauteng provincial legislature, an oversight body, initiated a meeting of the CHWs to learn about their challenges in providing community-based care under the national programme. All CHWs from the province were invited to the meeting, those who attended expressed their concerns over their yearly contracts, meagre remuneration, lack of working tools and poor relationships with facility-based staff: "*Mr Motsoaledi [Minister of Health] promised us in November that he is absorbing us. When we were called to Pretoria [National DoH] we thought that we were going to sign contracts as promised but he failed us*" (CHW, FGD, District C). The CHWs expressed their frustration at the lack of progress being made by the Department of Health to provide permanent employment. The legislature promised to attend to the CHWs grievances and provide feedback.

**Labour unions.**   Though the task team held numerous meetings with district management, they had limited negotiating powers, as they could not participate in the Department of Health Bargaining Council (DOHBC). The DOHBC is a formal structure consisting of registered unions, where workers' grievances (e.g., need for improved remuneration) are tabled for consideration by the employer. The task team needed representation in the DOHBC to escalate their demands. In 2018, the CHWs in District A joined the National, Education, Health and Allied Workers Union (NEHAWU), a registered labour union, to represent them in the DOHBC. The union was entrusted with negotiating for permanent employment, decent remuneration for the CHWs and other benefits associated with formal employment (e.g. leave allowance). One of the representatives of CHWs was also co-opted into the union's regional structures. Some CHWs did not join the union, as the decision appeared rushed, with minimal consultation. Others were concerned that they would not be able to afford the monthly premiums.

**Outcome of the task team and union mobilisations.**   When fieldwork ended, the task team and union had not been able to secure permanent employment for the CHWs. However, the union had negotiated with the Department of Health to increase the CHW stipend from R2 500 (136 USD) to R3 500 (192 USD) (Table 2). The CHWs felt motivated by the increment: "*The increase has motivated me because since we got the increase, I have been working harder than before*" (CHW, Interview, District A). Another CHW added with the increment they are able to afford necessities and uniform: "*Like now, there is a yellow T-shirt costing about R150, we are able to buy and pay for transport to come to work. On the other side, we buy groceries at home.*" (CHW, Interview, District A).

Although the CHWs were excited to receive the increment, they still wanted to be absorbed as permanent workers by the Department of Health. The union committed to continue engaging the Department on this issue and other concerns such as leave allowance.

In health facilities, the task team recorded some successes, a member of the task team commented: "*In facility X, the CHWs did not have access to a photocopy machine because they did not contribute to the facility budget to buy paper for the photocopy machine. . . how do you expect someone who is being underpaid to have money for such expenses? We fought for them to have access to the printer*" (CHW representative, Interview 2, District A).

Post the study and during the height of COVID-19 in 2020, when the need for motivated and effective CHWs became much more obvious to decision makers, the semi-urban-based teams received permanent employment with remuneration of between R9-11,000 (500–600 USD). The other CHW teams located in the rural areas did not receive this increment.

## Discussion

In the paper, we have reported the CHWs were poorly supervised, resourced and received meagre remuneration, their employment outsourced, without employment benefits and protection. In the semi-urban sites, CHWs established a task team to represent them but refused to join a formal union due to the fear that unions would push management insufficiently for their demands. The task team held regular meetings and led protests against clinic, district and provincial management to demand improved conditions of employment. After the recognition by the local provincial legislature, the task team joined a labour union (NEHAWU) in order to be able to participate in the local bargaining council. Though they were not successful in getting the government to provide permanent employment, the union negotiated an increase in stipend from R2 500 to R3 500. In contrast, in the rural sites, the CHWs were not actively demanding permanent employment due to their employment contracts being partly managed by NGOs; they were fearful of being recalled from the government programme.

The study findings are in line with the literature, studies in LMICs have shown many CHWs are functioning without the necessary resources and support [6]. The CHWs' lack of critical resources such as medicines, blood pressure and glucose machines make it difficult for them to provide the necessary care [3, 6]. The poor integration of CHW programmes into healthcare systems also demotivates the CHWs, as this means they often have to function without the backing of facility staff members [4, 6, 7]. The CHWs need health system support to provide comprehensive and quality health support to marginalised communities.

The study also reported that CHWs felt stuck as there were no career progression opportunities for them. The 2018 WHO guidelines for the optimisation of CHWs performance recommended countries make available career progression pathways for CHWs [6]. This hasn't been the case for CHWs in LMICs as many of them are on short-term contracts and paid a meagre stipend [7]. Investing in CHWs' professional development has short and long-term benefits. In the short term, CHWs work morale and motivation maybe enhanced, while in the long run their performance and credibility in the healthcare system and community may be boosted [6]. It is therefore crucial for countries like South Africa to prioritise upskilling CHWs with the intention of promoting them.

Since the end of data collection, the South African CHWs have engaged in further industrial action. In 2019, CHWs based in the Gauteng province led a protest against the Provincial Department of Health where they were demanding to be absorbed as permanent employees [23]. In 2021, the NEHAWU took the Department of Health to labour court to declare the 12 months contracts illegal and in contravention of the national labour laws, which state employment cannot be offered on a renewable basis for a period exceeding 3 months [24, 25]. The

court arbitrator ruled against the union citing a 2018 agreement signed by government and unions. The agreement refers to government dependence on an external grant to pay CHW current salaries, such that CHWs permanent employment is unsustainable. In the Mpumalanga province, there has been less activities by the CHWs to challenge the government to improve their conditions of employment.

Studies show the COVID-19 pandemic reinforced the need for stable CHW programmes [26–28]. In many countries, during the height of the pandemic, CHWs undertook case identification, participated in screening of people with COVID-19 symptoms, traced contacts and encouraged vaccine uptake [26–28]. However, poor access to working tools, supportive supervision and decent remuneration limited their effectiveness. A multi-country study conducted in Bangladesh, Pakistan, Sierra Leone, Kenya and Ethiopia showed CHWs were often expected to provide these services without personal protective equipment (PPE), and were unremunerated for the extra tasks they carried out during the pandemic [29]. In Brazil, at the height of COVID-19 infections, CHWs protested when they were expected to provide services while being underpaid, poorly trained and without PPE [30]. The Brazilian CHWs used social media and weekly webinars supported by labour unions to demand safer employment conditions. They took advantage of the vulnerabilities brought by the COVID-19 pandemic and exercised their agency and capacity to take strategic action [30]. This resulted in some municipalities beginning to purchase PPEs for CHWs undertaking household visits, while some invested in telemedicine to limit CHWs' direct contact with patients. As in Brazil, the pandemic in South Africa also highlighted the important role that CHWs play in reaching vulnerable groups and getting them to care [18, 31].

As highlighted by the current study, in many countries CHWs are without union representation to protect their labour rights [32, 33]. However, following the development of nationwide programmes in different countries, the shift from being disease-specific to comprehensive programmes [10], CHWs have begun to demand better employment conditions [11, 17]. In India, accredited social health activists' (ASHAs) and anganwadis'(These workers care for the health and wellbeing of women, children and other socioeconomically deprived groups.) unions have regularly led marches to demand formal employment with benefits [34]. Similar to South Africa, the unions have achieved small gains in terms of increased pay and social security benefits [34]. During the peak of the COVID pandemic, the Gauteng government offered the CHWs employment contracts similar to other permanently employed government staff members such as nurses [18, 31].

The study had several strengths and limitations. We spent 3 years collecting data in district A and this allowed us to familiarise ourselves with the history of the programme and conflicts between the CHWs and the Department of Health in the district. In the other districts, data collection was limited to three months. Due to our prolonged stay in District A, familiarity with the CHW programme and labour mobilisation, we used the site as the main comparator to the other sites. Second, the data collection tools were written in English, some participants preferred being interviewed in other local languages (e.g. IsiZulu and Sesotho). To achieve consistency of the translations, as part of pre fieldwork training, data collectors participated in role-plays to practice posing the same question in the different languages. This process allowed the data collectors to be comfortable and consistent in posing the interview questions.

## Conclusion and recommendations

Consistent mobilisation as demonstrated by the urban-based teams enabled the CHWs to successfully negotiate salary increments and advance their call for permanent employment.

However, in the rural sites, the CHWs were less able to join or establish labour representation due to fear of reprisal from NGOs management. It is important that CHWs are afforded the right to belong to a labour union in order to be able to negotiate for decent employment conditions. In optimising the motivation and performance of CHWs, the government need to prioritise the full integration of CHWs into the healthcare system, where they will be afforded their labour rights and support. This could include a provision of a decent salary with benefits such as leave and career progression opportunities for interested CHWs. It is further recommended that future studies prioritise the exploration of the CHWs conditions of employment post the COVID-19 pandemic. As previously stated, the pandemic highlighted the important role CHWs play in delivering healthcare to marginalised communities. Therefore it is important to assess whether their conditions have changed since the pandemic.

## Supporting information

**S1 Checklist. Consolidated Criteria for Reporting Qualitative research (COREQ) checklist.** (DOCX)

## Acknowledgments

We would like to thank the following individuals for their invaluable contribution: the data collectors, key informants, CHWs and their supervisors, health facility staff members and community representatives in the 4 districts. The support of the Sedibeng Health District management, in particular the former district director Mrs. Salamina Hlahane and community-based health services coordinator Mrs. Bridget Lefhoedi is highly appreciated.

## Author Contributions

**Conceptualization:** Hlologelo Malatji, Frances Griffiths, Jane Goudge.

**Data curation:** Hlologelo Malatji, Jane Goudge.

**Formal analysis:** Hlologelo Malatji, Jane Goudge.

**Funding acquisition:** Frances Griffiths, Jane Goudge.

**Investigation:** Hlologelo Malatji, Frances Griffiths, Jane Goudge.

**Methodology:** Hlologelo Malatji, Frances Griffiths, Jane Goudge.

**Project administration:** Hlologelo Malatji.

**Resources:** Hlologelo Malatji, Frances Griffiths, Jane Goudge.

**Software:** Jane Goudge.

**Supervision:** Hlologelo Malatji, Frances Griffiths, Jane Goudge.

**Validation:** Frances Griffiths, Jane Goudge.

**Writing – original draft:** Hlologelo Malatji, Jane Goudge.

**Writing – review & editing:** Hlologelo Malatji, Frances Griffiths, Jane Goudge.

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
