## [Decision Letter · Decision Letter 0]

15 Aug 2023

PGPH-D-23-01313

Mobilisation towards formal employment in the healthcare system: A qualitative study of community health workers in South Africa

Dear Dr. Malatji,

Thank you for submitting your manuscript to PLOS Global Public Health. After careful consideration, we feel that it has merit but does not fully meet PLOS Global Public Health’s publication criteria as it currently stands. Therefore, we invite you to submit a revised version of the manuscript that addresses the points raised during the review process.

This is a very interesting and important study examining the employment conditions and labor mobilization of community health workers (CHWs) in South Africa. The topic is highly relevant given the critical role CHWs play in delivering primary care, yet often face challenging working conditions.

The brief background on unions in South Africa is helpful for framing the later findings on CHW unionization attempts. The one area that could be strengthened is explicitly stating the study aim/objectives at the end of the introduction section. Currently the objectives are implicit but not directly stated

Some parts of the methods could be clarified or expanded. In particular, more details on the data analysis process, how themes were derived, and steps to ensure rigor would further strengthen this section.

For the data analysis, consider expanding on the coding process and how themes were derived from the codes. Did you use any specific qualitative analysis approach? How was rigor ensured (e.g., multiple coders, checking themes against raw data)?

We look forward to receiving your revised manuscript.

Kind regards,

Ejemai Eboreime, MD, MSc, PhD

Academic Editor

Journal Requirements:

1. We have noticed that you have uploaded Supporting Information files, but you have not included a list of legends. Please add a full list of legends for your Supporting Information files after the references list.

Reviewers' comments:

Reviewer's Responses to Questions

**Comments to the Author**

1. Does this manuscript meet PLOS Global Public Health’s publication criteria? Is the manuscript technically sound, and do the data support the conclusions? The manuscript must describe methodologically and ethically rigorous research with conclusions that are appropriately drawn based on the data presented.

Reviewer #1: Partly

Reviewer #2: Yes

2. Has the statistical analysis been performed appropriately and rigorously?

Reviewer #1: N/A

Reviewer #2: N/A

3. Have the authors made all data underlying the findings in their manuscript fully available (please refer to the Data Availability Statement at the start of the manuscript PDF file)?

Reviewer #1: Yes

Reviewer #2: Yes

4. Is the manuscript presented in an intelligible fashion and written in standard English?

Reviewer #1: Yes

Reviewer #2: Yes

5. Review Comments to the Author

Reviewer #1: The study aimed to examine CHWs' employment status and their actions towards formally integrating into the healthcare system in urban and rural health districts in South Africa, and analysed data from 2 other studies on the design and implementation of CHW programmes in South Africa. The authors reported the difficulties faced by the CHWs in different health districts, and described their mobilization efforts towards formalizing their employment in the healthcare system.

1. Methods:

a. The selection criteria for the participants for the individual interviews were not clear, especially for CHWs who were also the participants of the focus groups.

b. The distribution of the interviews among the different groups (e.g. CHWs, supervisors, facility staff, etc) was not indicated.

c. A more detailed description on how and where the observations were performed (e.g. health district) would add clarity.

2. Findings:

a. The planned analysis included a comparative analysis across "the different cases of context, processes and outcomes of mobilization", however these comparisons were not immediately evident in the findings. Perhaps presenting the results across cases in a table would demonstrate similarities and differences across the cases for both CHW employment conditions and CHW mobilization efforts (presented as Table 2) more clearly.

b. Please review total number of focus groups in Table 1. (Row 1 - total indicated is 9 but count is 10).

c. If possible, please include a description of the characteristics of the study participants, e.g. for CHWs, age, length of service; for key informants, specify who they were (e.g. supervisor, staff, etc).

d. I got the impression that the findings for part 2 on CHW mobilization came from only one health district, Sedibeng. If this is the case, then I suggest that this be stated explicitly in the findings. Also suggest to specify which interviewer group reported events/findings, i.e. CHW representatives, CHWs, supervisor, etc.

3. Discussion: The discussion focused mainly on developments in CHW actions during the COVID-19 pandemic, which, although important, was a different context from what was reported in the findings of the study. A broader discussion of the actions taken by CHWs to push for recognition of their integral role in the healthcare system and factors that have contributed to successful mobilizations/campaigns for better work conditions would have been appreciated.

Reviewer #2: Abstract

1. The methods sentences can use some detail thus:

a. Include sample size for each method

b. add a statement on analysis

2. The results part lacks explicit statements that align with the stated objectives “employment status, struggle for recognition and activities to establish labour representation” to tie down the stated findings. For example, what is the interpretation of the varying CHWs status in rural versus urban?

3. Strengthen the concluding statement.

Introduction

4. First paragraph reads too general and detached from the manuscript’s discourse, consider starting strongly with the problem which led to your study then narrowing down to what we know about the problem down to what is missing in what we know.

5. Align background aim with the abstract aim.

Methods

6. Consider adhering to some of the 21 Standards for Reporting Qualitative Research (SRQR).

7. Research design:

a. State the technical approach utilized in the study and justify its suitability to the study.

8. Study setting:

a. Consider using maximum variation technique as the criteria for selecting study setting. Add references to the study setting statements.

9. Sampling strategy: Consider having a separate section for sampling embedding it in the data collection section takes away from appreciating the sample population and technique.

10. Data analysis:

a. State data sources analyzed using thematic content analysis.

b. Consider stating how authors were involved in the process using their initials.

Results

11. The first section of the results:

a. lacks a general statement on the condition (or “status”) of the CHWs thus add as a sweeping statement in response to your initial objective.

b. Add an interpretative touch to your findings to ease the story articulation. For instance on the issue of career advancement the interpretation may read like “there is no advancement despite finishing training and being qualified”.

c. The impact of the conditions on effectiveness presented as they are seem as if they are diverting from the study focus on mobilization towards employment; consider framing them in terms of your other objective as part of activities to push change or the struggle itself.

d. Subthemes lack a storyline, revise to suit the broken-down aim in abstract.

12. Add an interpretive storyline to section two and link to gaps stipulated in background.

Discussion

13. First paragraph discussion seem like an expansion of results as it lacks references from other sources and comparison to other studies.

14. The discussion section need to be structured in short paragraphs and based on results. Considering the number of subthemes in the results section, the current three paragraphs do not add up.

Discussion

15. Add recommendations to the conclusion section.

6. PLOS authors have the option to publish the peer review history of their article (what does this mean?). If published, this will include your full peer review and any attached files.

**Do you want your identity to be public for this peer review?** For information about this choice, including consent withdrawal, please see our Privacy Policy.

Reviewer #1: No

Reviewer #2: No

---

## [Decision Letter · Decision Letter 1]

28 Dec 2023

PGPH-D-23-01313R1

Mobilisation towards formal employment in the healthcare system: A qualitative study of community health workers in South Africa

Dear Dr. Malatji,

Thank you for submitting your manuscript to PLOS Global Public Health. After careful consideration, we feel that it has merit but does not fully meet PLOS Global Public Health’s publication criteria as it currently stands. Therefore, we invite you to submit a revised version of the manuscript that addresses the points raised during the review process.

The manuscript is much improved. Please attach a completed COREQ checklist indicating how your manuscriptaligns with the reporting standards. Kindly see our editorial policy for guidance

Address any conflicts between the reviews so that it's clear which advice the authors should followProvide specific feedback from your evaluation of the manuscript

Please ensure that your decision is justified on PLOS Global Public Health’s publication criteria and not, for example, on novelty or perceived impact.

We look forward to receiving your revised manuscript.

Kind regards,

Ejemai Eboreime, MD, MSc, PhD

Academic Editor

Journal Requirements:

Additional Editor Comments (if provided):

Reviewers' comments:

Reviewer's Responses to Questions

**Comments to the Author**

1. If the authors have adequately addressed your comments raised in a previous round of review and you feel that this manuscript is now acceptable for publication, you may indicate that here to bypass the “Comments to the Author” section, enter your conflict of interest statement in the “Confidential to Editor” section, and submit your "Accept" recommendation.

Reviewer #1: All comments have been addressed

Reviewer #2: All comments have been addressed

2. Does this manuscript meet PLOS Global Public Health’s publication criteria? Is the manuscript technically sound, and do the data support the conclusions? The manuscript must describe methodologically and ethically rigorous research with conclusions that are appropriately drawn based on the data presented.

Reviewer #1: Yes

Reviewer #2: Yes

3. Has the statistical analysis been performed appropriately and rigorously?

Reviewer #1: N/A

Reviewer #2: N/A

4. Have the authors made all data underlying the findings in their manuscript fully available (please refer to the Data Availability Statement at the start of the manuscript PDF file)?

Reviewer #1: Yes

Reviewer #2: Yes

5. Is the manuscript presented in an intelligible fashion and written in standard English?

Reviewer #1: Yes

Reviewer #2: Yes

6. Review Comments to the Author

Reviewer #1: The article describes how consistent mobilization efforts among community health workers have led to improvements in their working conditions. The revised manuscript is much improved, more focused, and easier to read and understand.

Reviewer #2: PGPH-D-23-01313R1

Abstract

1. The methods section has been revised to include sample size and data analysis strategy.

2. The results section now aligns with the stated objectives “employment status, struggle for recognition and activities to establish labor representation”. However they need to be framed coherently;

• for instance, line 26 to 28 is a list of challenges which are expected to be unpacked as a trigger to protests but the sentence that follows also presents another trigger then jumps to industrial action aimed at improving employment condition “permanent employment and pay increment” which only appear in next sentence.

3. The benefits seem to be for semi-urban only, what of the rural? Include their fate to balance the conclusion.

Introduction

4. The background’s aim now aligns with the abstract’s aim.

Methods

5. The methods section has been well revised, it now meets acceptable standards for reporting qualitative research.

Results

6. The results have been improved but lack the aftermath version presented in the abstract. Decide whether you add document review (how you got the semi-urban news of improvements) as a new method to have full circle results.

Discussion

7. The discussion section has been revised to well specified thematic discussion paragraphs.

8. The discussion on the aftermath on line 583 to line 611 seems like a mix of results and discussion. It also lacks statements of other settings “Mpumalanga”, be it there is no information or such changes are yet to take effect.

Conclusion and Recommendations

9. A conclusion and recommendations have been added. Any recommendation for further study?

7. PLOS authors have the option to publish the peer review history of their article (what does this mean?). If published, this will include your full peer review and any attached files.

**Do you want your identity to be public for this peer review?** For information about this choice, including consent withdrawal, please see our Privacy Policy.

Reviewer #1: No

Reviewer #2: No

---

## [Editor Report · Decision Letter 2]

9 Feb 2024

Mobilisation towards formal employment in the healthcare system: A qualitative study of community health workers in South Africa

PGPH-D-23-01313R2

Dear Mr Malatji,

We are pleased to inform you that your manuscript 'Mobilisation towards formal employment in the healthcare system: A qualitative study of community health workers in South Africa' has been provisionally accepted for publication in PLOS Global Public Health.

Best regards,

Ejemai Eboreime, MD, MSc, PhD

Academic Editor